# Prion shedding is reduced by chronic wasting disease vaccination

**Hanaa Ahmed-Hassan**[1,2,3,4], **Dalia Abdelaziz**[1,3,4], **Yo-Ching Cheng**[1,3,4], **Kevin Low**[1,3,4], **Shirley Phan**[1,3,4], **Byron Kruger**[1,3,4], **Chimoné S. Dalton**[1,3,4], **Lech Kaczmarczyk**[5], **Walker S. Jackson** [5], **Sabine Gilch**[1,3,4], **Hermann M. Schätzl**[1,3,4]*

1 Calgary Prion Research Unit, Faculty of Veterinary Medicine, University of Calgary, Calgary, Alberta, Canada, 2 Zoonoses Department, Faculty of Veterinary Medicine, Cairo University, Giza, Egypt, 3 Hotchkiss Brain Institute, Cumming School of Medicine, University of Calgary, Calgary, Alberta, Canada, 4 Snyder Institute for Chronic Diseases, University of Calgary, Calgary, Canada, 5 Linköping University, Linköping, Sweden

* hschaetz@ucalgary.ca

## Abstract

Chronic wasting disease (CWD) is a strictly fatal and highly contagious prion disease of wild and farmed cervids currently expanding in North America. Prion diseases are caused by conversion of the cellular prion protein to its pathological isoform PrP^Sc. Vaccination is considered a promising strategy to contain CWD, even though prion diseases do not show classical immune responses. For CWD containment, it is important that vaccines reduce shedding of prions in excreta, a major contributor to transmission. Here, we tested the effect of vaccines on prion shedding in feces and urine by vaccinating and prion infecting knock-in mice that recapitulate CWD pathogenesis as found in cervids. Vaccination reduced or even prevented CWD shedding in feces and urine collected between 30–90% of incubation time to disease. This is the first report showing that prion shedding can be blocked in a prion disease. For CWD specifically it may reduce the environmental prion burden and break the disease transmission cycle.

## Author summary

Chronic wasting disease (CWD) is an infectious prion disease of free-ranging and captive deer and elk that is always deadly and currently expanding in North America. Infected cervids shed infectious CWD prions continuously into the environment via urine and feces long before animals develop clinical signs after around 2 years. There, prions remain infectious for a long time and contribute to the very efficient and hard to control horizontal transmission of CWD. Vaccination recently emerged as a promising tool that could help to contain CWD. Given the environmental component of CWD vaccines ideally also should reduce the shedding of prions into the environment. We addressed this important aspect in a mouse

**Data availability statement:** All data underlying our findings are fully available, without restrictions, and within this manuscript or uploaded as Supporting information.

**Funding:** We acknowledge funding for this research from the Natural Sciences and Engineering Research Council of Canada (NSERC Alliance Grant ALLRP 571218-21 to SG and HMS). We are grateful for financial support from Alberta Innovates (201600023; 222300851 to HMS), the National Institutes of Health (NIH R01AI156037 to HMS), NSERC (RGPIN-2020-04581 to HMS), Results Driven Agriculture Research (RDAR 2025T3838R to SG and HMS) and Alberta Environment and Protected Areas (128153289 to SG and HMS). The funders had no role in study design, data collection and analysis, decision to publish, or preparation of the manuscript.

**Competing interests:** The authors have declared that no competing interests exist.

model that recapitulates CWD pathogenesis as found in cervids. Importantly, when compared to unvaccinated controls, we found that vaccination decreases CWD shedding in feces and urine at the examined preclinical stages representing 30–90% of incubation time to disease. Taken together, our vaccination strategy seems to provide two additive effects. It improves individual survival that hopefully will translate into a positive population effect and it reduces prion shedding, likely translating into a reduction over time of CWD prions in the environment.

## Introduction

Prion diseases or transmissible spongiform encephalopathies (TSE's) are fatal infectious neurodegenerative disorders of the central nervous system (CNS). These diseases are caused by the conversion of the normal cellular prion protein (PrP$^C$) into the misfolded and pathological scrapie isoform termed PrP$^{Sc}$ [1]. Accumulation of PrP$^{Sc}$ leads to spongiform degeneration and neuronal loss in the CNS [2]. Chronic wasting disease (CWD) is considered the most contagious prion disease in animals, affecting both wild and captive cervid populations such as elk, deer, reindeer and moose [3–5]. CWD is expanding rapidly in North America, and by April 2025 it has been detected in 36 U.S. states and 5 Canadian provinces [3]. CWD has also been found in South Korea and since 2016 in Norway, Finland, Sweden [3,6].

CWD is the only known prion disease that affects wildlife, which is a significant challenge for controlling the disease. Prion infectivity is present in the CNS as well as many peripheral tissues, body fluids and secretions of CWD infected animals [7,8]. Transmission of CWD occurs directly or indirectly between animals, due to shedding of CWD prions through feces, saliva and urine, which contaminates the environment. Importantly, shedding starts early in pre-clinical disease stages [9–12]. Environmental contamination of CWD persists for years and serves as a reservoir for infecting the same species, or it may facilitate transmission to other susceptible species. CWD is economically relevant and can directly impact hunting and tourism industries, and potentially food safety and food security of people who rely on subsistence hunting and the consumption of venison. The uncertain zoonotic potential of CWD remains a significant risk for humans [13]. CWD can be transmitted experimentally to non-cervid hosts including voles, hamsters, ferrets, sheep, cats, mink, pigs and cattle [14–18]. Several studies that examined the zoonotic potential of CWD concluded that the risk is very low [19–24]. However, a recent study showed that CWD can be transmitted to mice expressing the human prion protein, with CWD infectivity found in brain and feces, the latter if it happens, making feces a source of infection between humans [20]. In addition, CWD prions have been detected in skeletal muscle [25] and antler velvet of infected cervids, which raises the possibility of zoonotic transmission through consumption of contaminated venison and/or utilization of infected cervid products in Asian medicines [26].

To date, there is no effective treatment or vaccination available against CWD. Prion disease does not evoke a detectable immune reaction in the host, as the endogenous host PrP$^C$, a self-protein, is converted into PrP$^{Sc}$ when prions start to

replicate. The main obstacle to having a vaccine against any prion disease is therefore overcoming the self-tolerance without inducing unwanted side effects. Conceptually there are two main targets for active vaccination against prion diseases: targeting PrP^C or PrP^Sc [5,27–30]. The approach pioneered by our group targets PrP^C (termed 'PrP^C vaccination') mostly induces self-antibodies that bind to surface-located PrP^C and interfere with cellular prion replication [31–33]. Ideally, an effective vaccine for CWD should decrease the number of infected cervids over time, reduce the shedding of already infected cervids, and reduce the contamination of the environment.

Our previous studies have shown that vaccination targeting PrP^C overcame self-tolerance in cervids and transgenic mice, produced detectable humoral and cellular immune responses, and increased the survival time in mouse models of CWD without inducing unwanted side effects [5,34–36]. No study has yet tested the efficacy of active vaccination on CWD prion shedding. In transgenic mouse models of CWD infection this is not possible to test as they apparently do not shed prions into urine and feces and often do not have a complete attack rate when challenged peripherally with prions [37,38]. In the current study, we show the effect of vaccination on CWD shedding in feces and urine, using cervidized knock-in (KI) mice that recapitulate CWD pathogenesis as found in the cervid host including prion shedding [37,38]. We used recombinant dimeric deer prion protein (Ddi) and monomeric mouse prion protein (Mmo), vaccines we have shown to produce humoral immune responses against PrP^C [5,31,33], and able to protect vaccinated cervidized mice against prion challenge [34–36]. To investigate the effect of Ddi and Mmo vaccination on CWD shedding, feces and urine were collected every 50 days post infection (dpi). Fecal samples contain inhibitors that make CWD detection more difficult by in vitro amplification assays such as protein misfolding cyclic amplification (PMCA) and real-time quaking-induced conversion (RT-QuIC) [39–42]. To improve the sensitivity of those assays, iron oxide magnetic extraction (IOME) beads have been used [43,44]. In the current study, we combine IOME with PMCA followed by RT-QuIC to increase sensitivity and specificity of the assay. The combination of the three techniques is referred here as 'IPR technique'. Our study demonstrates that vaccination targeting PrP^C leads to a decrease in CWD shedding in feces and urine at preclinical stages in a CWD mouse model. This vaccination strategy therefore has two additive effects: it improves individual survival which will translate into population effect, and it reduces prion shedding translating into reduction of CWD prions in the environment.

## Results

### Vaccination targeting PrP^C delays CWD neuro-invasion in knock-in mice inoculated with CWD

In this study, we tested the effect of PrP^C vaccination on CWD prion shedding in feces and urine using a cervidized KI mouse model. We used gene-targeted KI mice where mouse PrP was replaced with cervid PrP, which recapitulate the pathogenesis of CWD, including shedding of CWD prions into feces and urine [37,38]. For vaccination, we used three groups of KI mice (n = 8/group): two vaccine groups receiving either recombinant dimeric deer (Ddi) or monomeric mouse (Mmo) PrP with CpG as adjuvant, or CpG adjuvant alone as control group. All mice received one priming dose (100 ug of the protein plus 5 µM of CpG) and four booster doses subcutaneously at three weeks intervals (s.c.) (50 ug of the protein plus 5 µM of CpG). The control group received 5 µM CpG in PBS only. CWD prion infection was performed intraperitoneally (i.p.) using brain homogenate of KI-mouse adapted reindeer CWD (Fig 1A). This mouse model was expected to develop clinical signs of prion infection between 450 and 500 dpi [37]. We harvested various mice before reaching the clinical endpoint, and used such pre-clinical mice to assess differences in PrP^Sc levels in brain and spinal cord between immunized and control mice by immunoblot analysis. Overall, PrP^Sc levels in brain (Fig 1C and 1D) and spinal cord (Fig 1E and 1F) of mice with comparable incubation time in the Ddi or Mmo vaccinated groups were reduced compared to control animals. As expected, PrP^Sc levels in spinal cord were higher than in brain, reflecting the peripheral mode of infection and the process of neuro-invasion (Fig 1G-1I). Only five mice reached terminal prion disease and presented clinical signs such as belly movement, extension, and clasping of hind limbs. One CpG (#6) mouse showed clinical signs of terminal prion disease at 462 dpi, while two mice from the Mmo (#2 and 4) vaccinated group developed terminal prion disease at 494 and 495 dpi, and two mice in the Ddi (#2 and 3) vaccinated group at 488 and 495 dpi, respectively. PrP^C vaccination

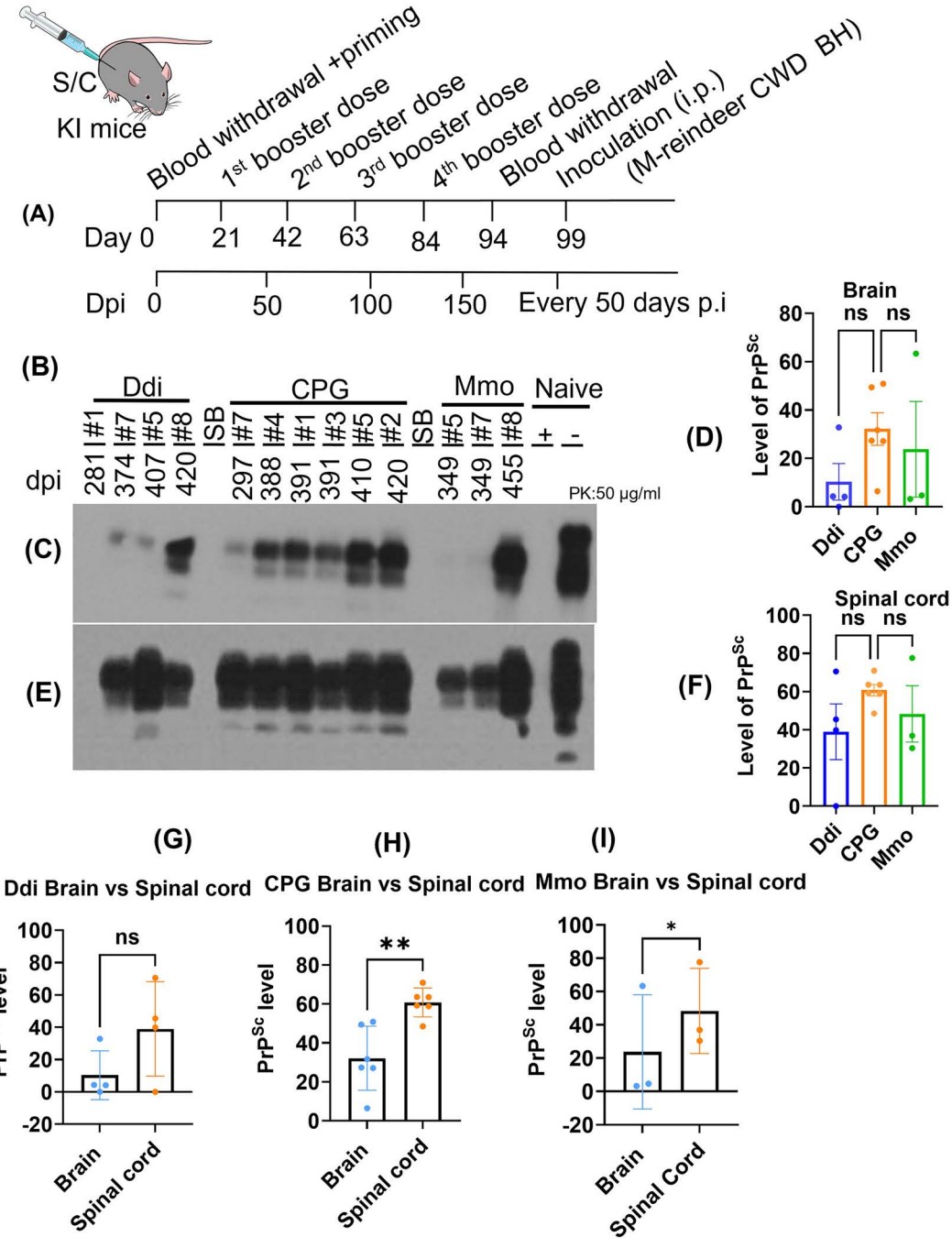

**Fig 1. Immunoblot showing PrPSc levels in brain and spinal cord of cervid KI mice. (A)** Schematic diagram showing the study design for vaccination and **(B)**, the time points of feces and urine collection. Ddi vaccinated group (n = 8), CPG control group (n = 8) and Mmo vaccinated group (n = 8) were inoculated intraperitoneally with 1% brain homogenate (BH) of mouse-adapted reindeer CWD. Samples were digested with 50 µg/ml PK and loaded onto SDS-PAGE as labelled at the top of the blot, with **(C)** showing PrPSc levels in brain and **(E)** in spinal cord. Membranes were probed with the anti-PrP mAb 4H1 (1:500). Naïve depicts brain homogenate of non-inoculated KI mice. The PrPSc levels in brain **(C)** and spinal cord **(E)** were densitometrically quantified for the three vaccination groups (**(D)** brain, **(E)** spinal cord), the graphs in **(G), (H)** and **(I)** compare brain signals to spinal cord for the three vaccination groups (Ddi, CpG and Mmo). SB stands for sample buffer and denotes a free lane. Graphs were generated by GraphPad Prism (version 10). Statistical analysis was done using paired-t test. ns = not significant, * p-value = 0.0440 and ** p-value = 0.0038.

increased survival time by around 30 days compared to the control group,. These data show that vaccination delayed the process of neuro-invasion and extended the time to clinical disease in both vaccinated groups compared to the control group.

### PrP$^c$ vaccination induces a humoral immune response in cervid PrP expressing KI mice

For testing specific humoral immune responses, we analyzed post-immune sera from all mice in ELISA, using plates coated with Ddi as antigen (Fig 2A and 2B) as well as plates coated with Mmo (Fig 2C and 2D). We found that all mice vaccinated with Ddi immunogen showed reactivity against Ddi antigen (**Fig 2A and 2B**), but not against Mmo (**Fig 2C and 2D**) at a 1:100 dilution. Mice vaccinated with Mmo immunogen reacted well in both situations. The endpoint dilution of sera showed that most sera were still reactive at a 1:30,000 dilution (Fig 2E). These data show that both immunogens break the self-tolerance to PrP and produce high antibody titers.

To gain a deeper understanding of the specificity of the immune responses generated by the vaccine candidates, we performed a linear epitope mapping analysis, covering full-length mature cervid PrP. The post-immune sera used in this analysis were average from 4 mice in the Ddi-vaccinated group (mice # 1, 3, 4 and 6) and 4 mice from the Mmo group (1, 2, 3 and 6), so it reflects average reactivity in each group. Of note, there was no reactivity to the polyhistidine tag (poly-his epitope) and the linker sequence (S1 Fig). Interestingly, both immunogens resulted in a similar reactivity to the linear

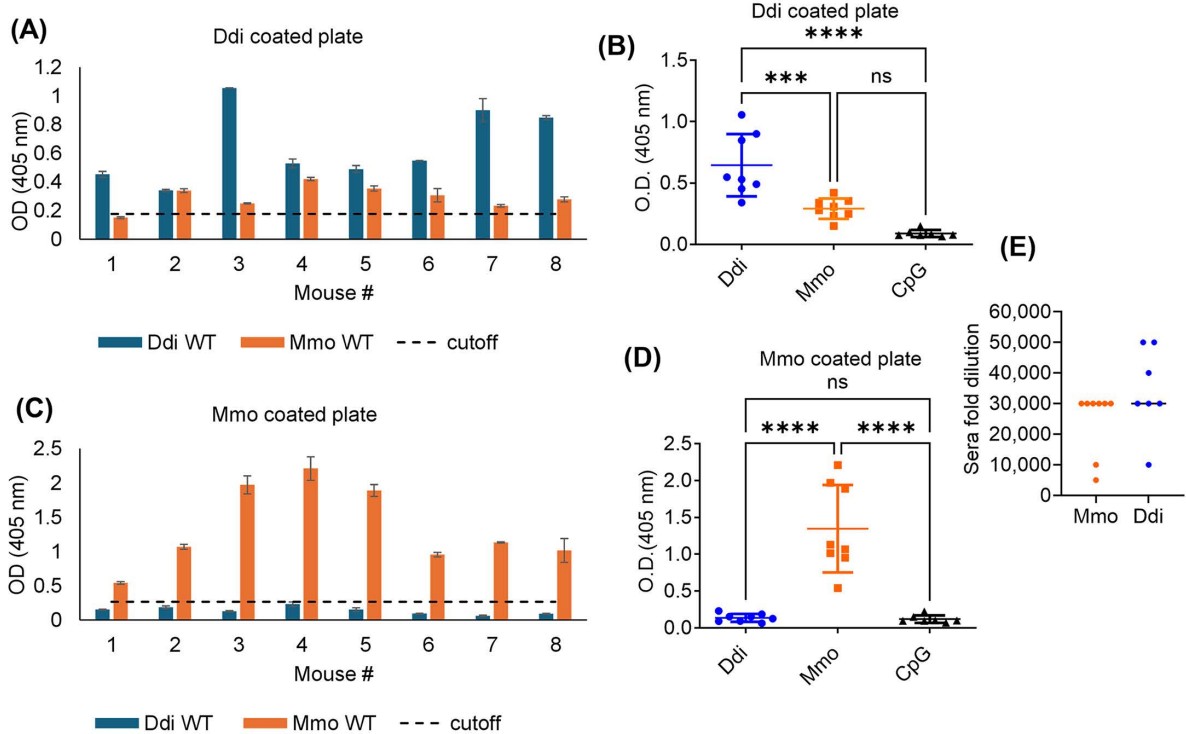

**Fig 2. Humoral immune response of Ddi and Mmo vaccinated KI mice to immunogens.** ELISA plates were coated with Ddi immunogen (**A, B**) or Mmo immunogen **(C, D)**. Sera were isolated after the 4$^{th}$ dose of vaccination (n = 8 per vaccination group.) All sera were diluted at 1:100, CpG mice sera were used as a negative control and GAM was used as secondary antibody at 1:4,000. One-way ANOVA was done followed by Tukey's test. Statistical significance = **** p-value <0.0001, *** p-value = 0.0007. **(E)** End-point ELISA antibody dilutions are shown for the two vaccinated groups. Each mouse antibody titer was determined by endpoint dilution and is represented by a data point. The y-axis represents the serum fold dilution, and the x-axis represents the treatment groups. The dashed horizontal line represents the cut-off, which is the average OD of CpG sera + 5 x SD.

epitopes, with exception of epitope #11 (sequence: NTFVHDCVNITVKQHTVTTTT) (**S1 Table**), to which Ddi-vaccinated sera reacted but not Mmo sera (S1 Fig).

## Vaccination decreases CWD prion shedding in feces of cervidized KI mice

To test the effect of vaccination on shedding of CWD prions in feces, individual and pooled fecal samples were collected and analyzed starting at 200 dpi. Fecal samples were subjected to homogenization and iron oxide magnetic extraction (IOME), followed by protein misfolding cyclic amplification (PMCA) and real-time quaking-induced conversion (RT-QuIC), summarized here as IPR readout. Of note, this is the first description of IPR to detect CWD in mouse feces. First, we started screening pooled fecal samples taken at 350 dpi. IPR, with RT-QuIC as final readout, showed that the seeding activity in the Ddi group was lower than that of the CpG and Mmo groups. Pooled fecal samples from one cage with Ddi-vaccinated mice showed no reactivity (**Fig 3A**), whereas pooled samples from the second cage had reactivity, but lower than that of CpG control mice or Mmo-vaccinated mice (**Fig 3B**). To get a better resolution of this finding in time and at the individual mouse level, we next studied fecal samples of individual mice. For this, we analyzed samples taken at 200, 250, 300 and 350 dpi, representing 45%, 55%, 65% and 75% of the incubation time. We observed that CWD shedding into feces can be detected at as early as 200 dpi in KI mice. Interestingly, we found that only 50% fecal samples from Ddi-vaccinated mice at 200 dpi tested positive, whereas samples from Mmo-vaccinated mice were positive at 87.5%, and samples of CPG control mice at 100% (Fig 4A- 4D and Table 1). In addition, samples from Ddi- and Mmo-vaccinated mice still testing positive showed overall a lower seeding activity, as determined by a higher time to threshold (Fig 4E), lower maximum of range (Fig 4F), and lower area under the curve (Fig 4G) analysis. These data show that vaccination with Ddi and Mmo immunogens reduced shedding of CWD prions into feces at 200 dpi. Similar results were obtained at 250, 300 and 350 dpi. At 250 dpi, four out of seven mice (57%) tested positive for shedding in the control group versus two out of seven (28%) in the Ddi-immunized group (S2 Fig and Table 1). At 300 dpi, a similar trend was found, with two out of five mice (40%) testing positive in Ddi-vaccinated mice, but 100% positivity in the CpG control group (S3 Fig and Table 1). Fecal samples of Ddi-vaccinated mice that were still positive had lower quantities of CWD prions in the feces taken at 200, 250 and 300 dpi (Figs 4, S2 and S3E-S3G), with a similar trend for fecal samples from Mmo-vaccinated mice. Interestingly, at 350 dpi, the positivity rate was lowest in the Mmo-vaccinated group (20%), compared to Ddi-vaccinated mice (66%) CpG-only treated control mice (71%) (S4A-S4D Fig and Table 1). As before, samples still being reactive showed less signal intensity when analyzing time to threshold, area under the curve or maximum range compared to CpG-only treated mice, now more pronounced for Mmo-vaccinted mice (S4E-S4G Fig).

When correlating shedding results with antibody titers, the Ddi sera of mice #6, 7 and 8 showed the highest antibody titers, with 1:40,000 and 1:50,000, respectively. Ddi mice #7 and 8 showed shedding activity only at one of the four time points analyzed (S3 Table). In addition, Ddi mice #7 and 8 showed low PrP$^{Sc}$ levels in spinal cord. On the contrary side, Mmo mouse #8 has a low antibody titer (1:5,000) and high PrP$^{Sc}$ levels in brain and spinal cord, when compared to Mmo mice #5 and 7 with higher titers (1:30,000) (**Fig 1C** and **1E** and **S3 Table**).

Taken together, fecal samples from individual Ddi- and Mmo-vaccinated mice showed lower positivity rates than samples from CpG control mice, at 200, 250, 300 and 350 dpi (**Table 1**). In addition, CpG samples consistently showed the highest prion seeding activities compared to the vaccinated samples. This shows that vaccination consistently reduced shedding of CWD prions into feces, resulting in less positive fecal samples and in lower CWD prion amounts in still positive samples.

## Vaccination with Ddi immunogen blocks shedding of CWD prions in urine of cervidized KI mice

Having seen a consistent reduction of CWD prion shedding in the feces of vaccinated mice, we next wanted to see whether shedding is also reduced in the urine of vaccinated mice at various preclinical stages. To complement our shedding analysis in feces, which analyzed samples taken between 200 and 350 dpi only, we chose for the urine analysis also

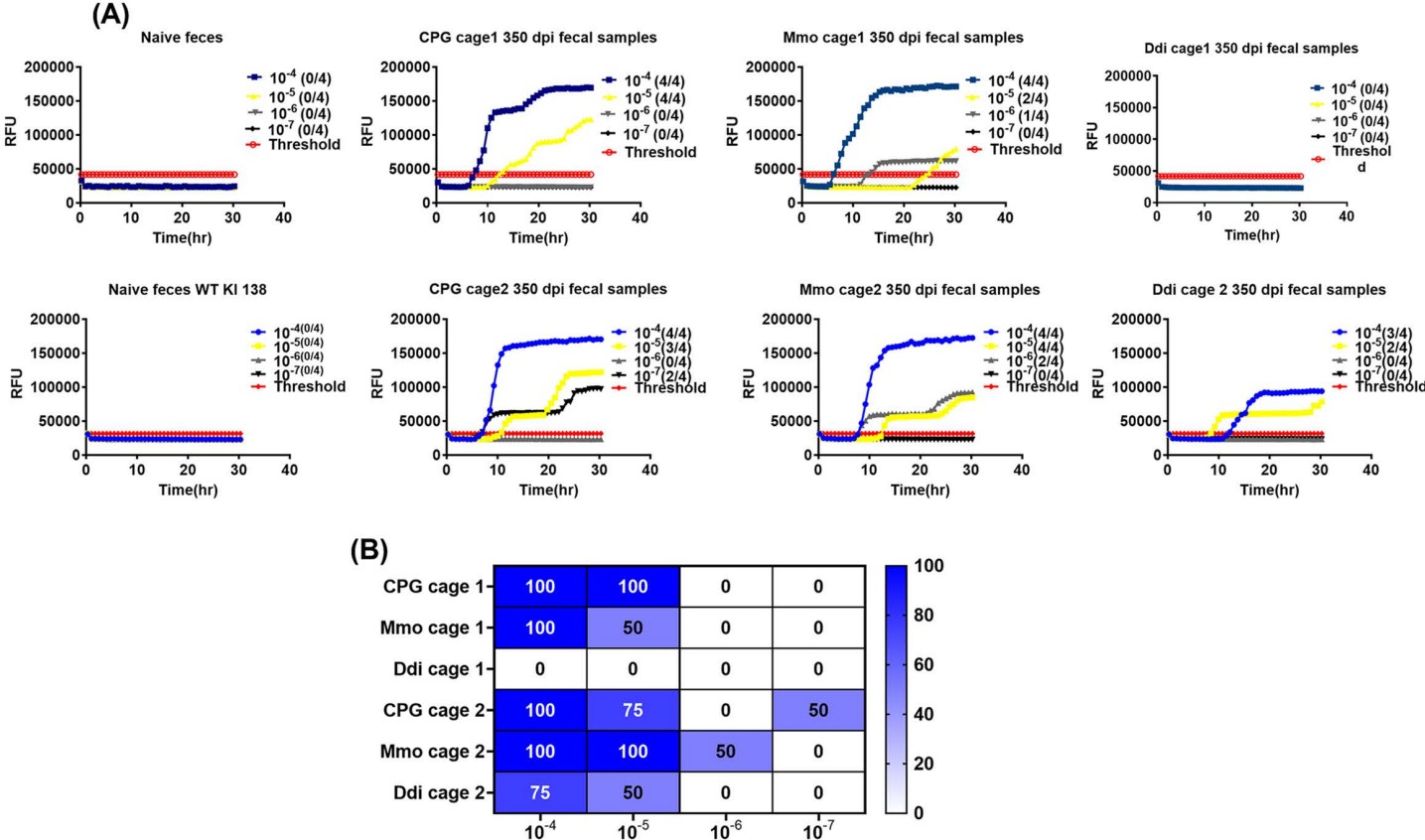

**Fig 3. RT-QuIC data for the PMCA products showing the difference in the CWD seeding activity between KI vaccinated and control groups at 350 dpi.** Ten % pooled fecal homogenate from all groups (two cages per group) was extracted using IOME followed by three rounds of PMCA reactions using KI wildtype cervid PrP$^C$ substrate seeded with 1:10 dilution of the fecal homogenate, in duplicates for all vaccinated and control groups. Positive control for the PMCA was naïve feces spiked with mouse-adapted CWD reindeer; negative control was naïve feces. Samples and controls were subjected to 3 rounds of PMCA. The PMCA products were analyzed using RT-QuIC assay at $10^{-4}$ to $10^{-7}$ dilution. **(A)** Representative RT-QuIC graphs showing the seeding activity in feces from KI mice vaccinated subcutaneous with Ddi or Mmo, or injected with CPG as control, followed by intraperitoneally inoculation with mouse-adapted CWD. Samples were considered positive when 2 out of 4 wells crossed the threshold, defined as the average relative fluorescent unit (RFU) of the negative control group plus five times its standard deviation. The y-axis represents the RFU, and the x-axis represents the time in hours (hr). **(B)** Summary of the RT-QuIC analysis of prion seeding in the 350 dpi pooled feces of KI mice. The heat map indicates the percentage of positive RT-QuIC replicates out of the total of four replicates analyzed. The scale ranges from 0 (all replicates were negative) to 100 (all replicates were positive). Graphs were generated using GraphPad Prism (version 10).

earlier (150 dpi, corresponding to 30% of incubation time) and later samples (450 dpi, corresponding to 90% of incubation time). Since it was technically not possible to analyze urine samples from individual mice due to the small volumes available, we decided to pool urine samples. Urine samples were analyzed using IOME extraction followed by three rounds of PMCA, or the IPR technique as described above. Remarkably, Ddi and Mmo-immunized mice did not shed CWD prions in the urine at 150, 250 and 450 dpi, when coupling IOME extraction and PMCA with immunoblot read-out (Fig 5A and 5B). CWD seeding activity was also not detected in urine samples of Ddi and Mmo-vaccinated mice at 150 and 250 dpi when performing the more sensitive IPR analysis as revealed in the RT-QuIC heat map (**Fig 5C and 5D**). At both time points, there was no detectable reactivity in urine samples of vaccinated mice, with an apparently 4-log higher conversion activity in urine samples of CpG control mice, where positivity was found at $10^{-1}$ to $10^{-4}$ dilutions of RT-QuIC reactions of the IPR read-out. These results indicate the presence of prions in urine of control mice, which was strongly reduced in vaccinated

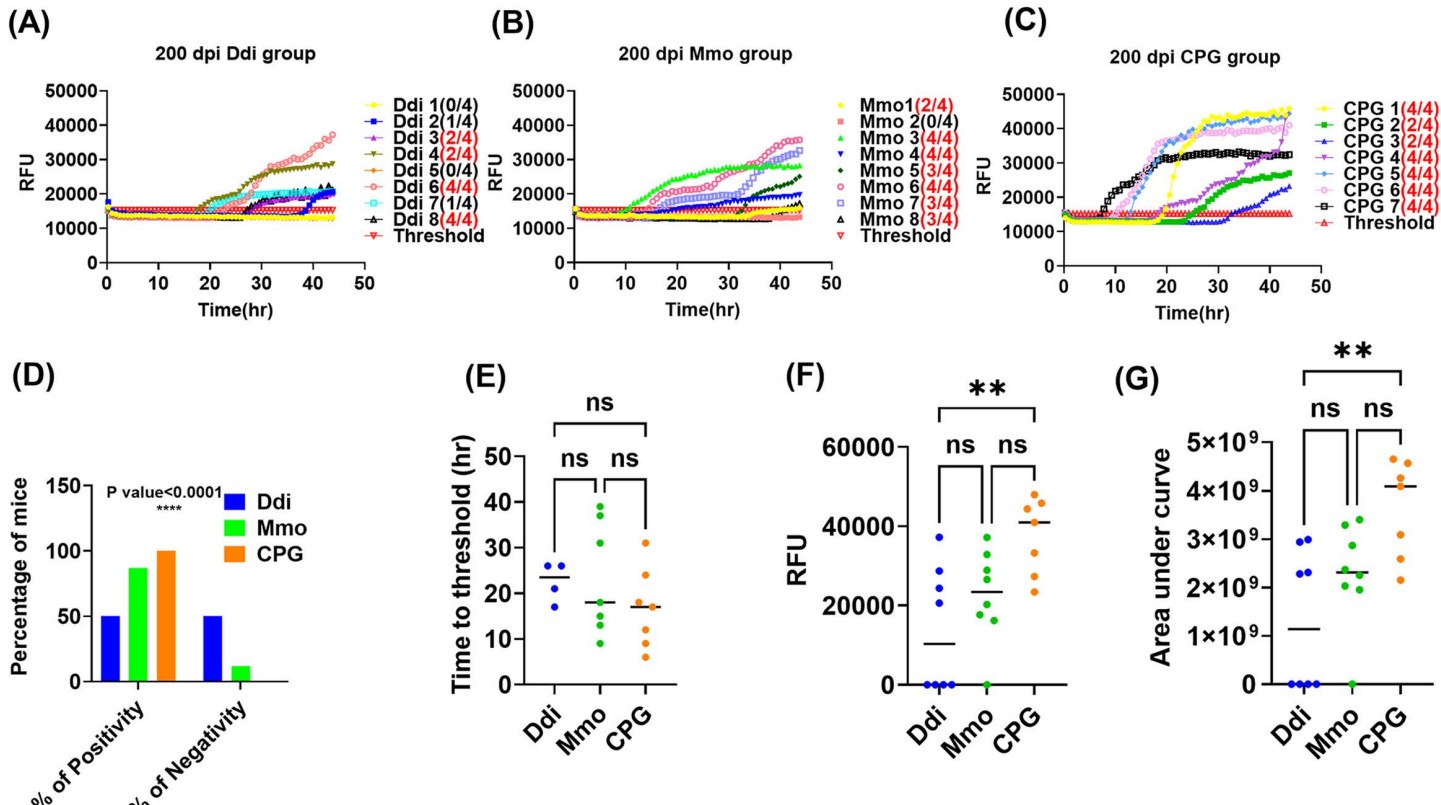

**Fig 4. Seeding activity in individual fecal samples of KI vaccinated and control group at 200 dpi.** Three rounds of PMCA reactions were done using KI substrate seeded with individual $10^{-1}$ dilution of 200 dpi 10% fecal homogenate samples after IOME, in duplicates for all vaccinated and control groups. Positive control for the PMCA was naïve feces spiked 1:100 with mouse-adapted CWD and negative control was naïve feces subjected to three cycles of PMCA reaction. The PMCA products were analyzed using RT-QuIC assay at $10^{-1}$ dilution. **(A-C),** representative RT-QuIC graphs showing the seeding activity in feces from KI mice vaccinated with Ddi or Mmo and CPG control group. Samples were considered positive when 2 out of 4 wells crossed the threshold, which defined as the average RFU of the negative control group plus five times its standard deviation. The y-axis represents the RFU, and the x-axis represents the time in hours (hr). **(D),** Chi square test, **(E),** time to threshold, **(F),** maximum of range and **(G),** area under curve. Graphs were generated using GraphPad Prism (version 10). Statistical analysis was done using Chi-square test **(D)** and **** p-value < 0.0001, or One-way ANOVA followed by a Tukey's multiple comparison. For maximum of range **(F)** ** p-value = 0.0046, for area under curve **(G)** ** p-value = 0.0034. Data in **(E)** were not significant. ns: not significant.

**Table 1. Difference in the CWD shedding in feces between KI vaccinated and control groups at different time points.**

|  |  | %of positivity |  |  |
| --- | --- | --- | --- | --- |
| Dpi (Dilution) | 200 dpi | 250 dpi | 300 dpi | 350 dpi |
| Ddi | (4/8) 50% | (2/7) 28.6% | (2/5) 40% | (4/6) 66.6% |
| Mmo | (7/8) 87.5% | (4/7) 57.1% | (5/6) 83.3% | (1/5) 20% |
| CPG | (7/7) 100% | (4/7) 57.1% | (5/5) 100% | (5/7) 71.4% |

mice (**Figs 5C**, **5D** and **S5**). At 450 dpi, we had samples from 6 pre-clinical mice in total only (**S2 Table**). Samples from the 3 remaining Ddi-vaccinated mice (#2–4) were negative in all RT-QuIC dilutions, indicating no detectable shedding in urine. Samples from Mmo-vaccinated mice (n = 2, #2 and 4) were positive up to the $10^{-4}$ dilution. Samples from the remaining

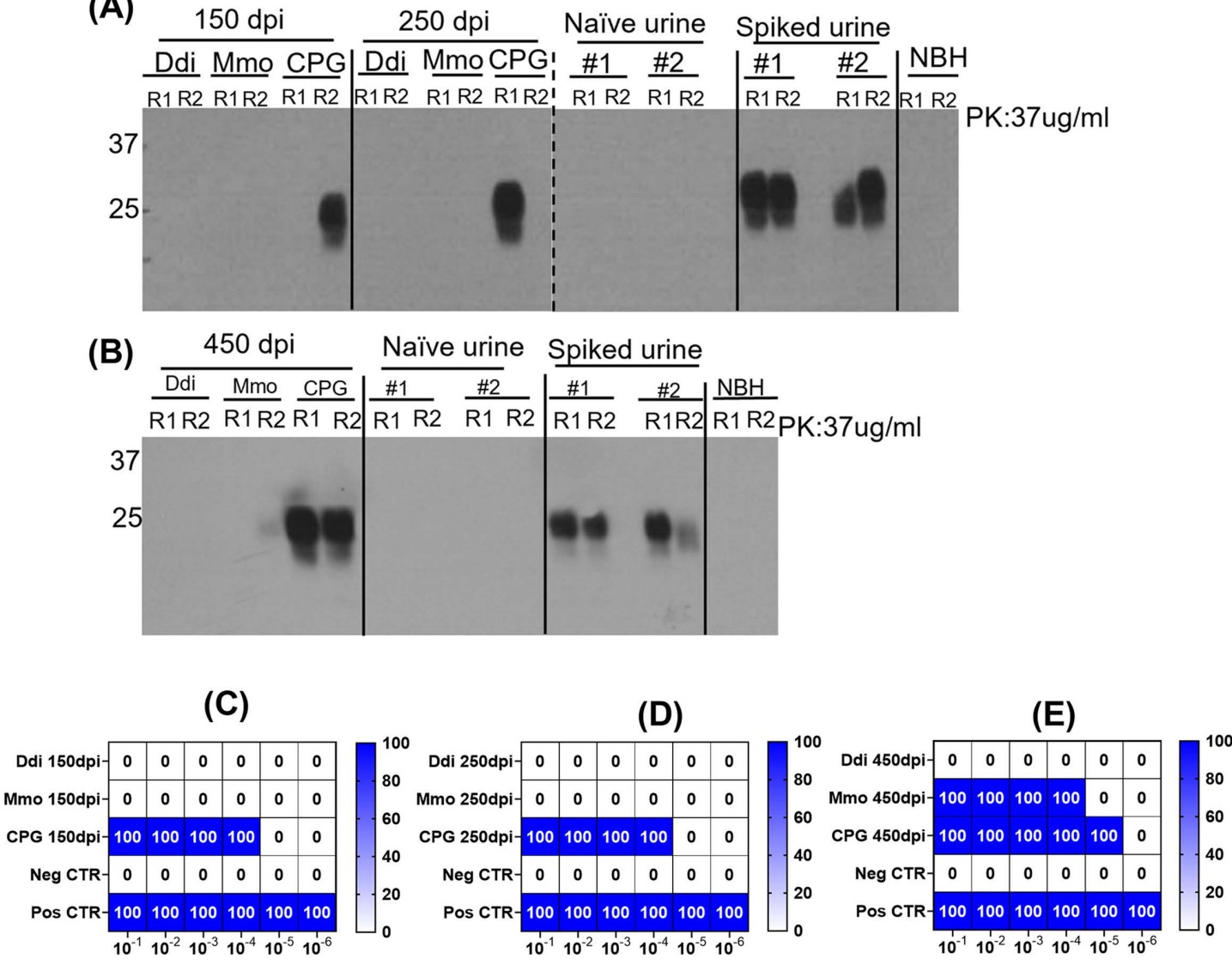

**Fig 5. Representative Western blots showing PrPSc levels in third round of PMCA reactions seeded with urine of Ddi, Mmo vaccinated or CPG control KI mice in duplicates at different time points. (A)**, 150 and 250 dpi and **(B)**, 450 dpi samples were subjected to IOME followed by three rounds of PMCA. Negative control for the PMCA is naïve urine and positive control for the PMCA was naïve urine spiked (1:100) with brain homogenate of mouse-adapted CWD subjected to PMCA reaction. PMCA products were digested for 1 hour with 37 µg/ml PK at 37°C and were probed with anti-PrP mAb 4H11 (1:500). NBH: normal brain homogenate, used as negative PMCA control. **(C-E)**, summary of the RT-QuIC analysis of prion seeding in the pooled urine of KI mice at **(C)** 150 dpi, **(D)** 250 dpi, and **(E)** 450 dpi. The heat map indicates the percentage of positive RT-QuIC replicates out of the total of four replicates analyzed. The scale ranges from 0 (all replicates were negative) to 100 (all replicates were positive). Graphs were generated using GraphPad Prism (version 10).

CpG control mouse (#6) tested positive up to the $10^{-5}$ dilution (**Figs 5E** and **S6**), indicating a 5-log reduction in the IPR readout in Ddi-vaccinated mice.

Taken together these results indicate that vaccination with the Ddi immunogen can reduce or even prevent shedding of CWD prions in urine in the KI mouse model of CWD infection.

**Reduction of prion shedding is not caused by anti-PrP antibodies that interfere with the IPR read-out**

We were wondering whether anti-PrP antibodies if present in feces and urine would interfere with the IPR readout. This is an important consideration as such antibody interference would result in false negative results in vaccinated animals and question the above observed reduction of prion shedding. To address this, we performed spiking experiments where naïve fecal samples were pre-treated with defined amounts of antibodies and CWD prions before performing the IPR read-out. A 10% fecal homogenate derived from naïve KI mice was incubated with pre-immune or post-immune sera from vaccinated KI mice described here, or with a 1:500 dilution of the anti-PrP monoclonal antibody 4H11 or the polyclonal antibody 142 for 90 minutes followed by spiking with brain homogenate (1:100 final dilution) of mouse adapted CWD, or left without spiking ('naïve'). All samples were subjected to the IPR readout exactly as described above. In all situations no effect of antibody presence was detected on either the PrPSc signals in immunoblot after PMCA (**Fig 6A**) or the seeding activity in RT-QuIC upon PMCA (**Fig 6B**). Importantly, this was also not the case when higher amounts of anti-PrP antibodies were present, when pre-treated with the high concentration experimental anti-PrP monoclonal or polyclonal antibodies. This data proves that anti-PrP antibodies do not interfere with the IPR readout.

In addition, we tested whether we can detect antibodies against the immunogens in feces and urine using ELISA (**S8 Fig**). Feces from Ddi-vaccinated mice (300 dpi) were diluted 1:50 (w/v) (**S8A Fig**) or 1:10 for urine (**S8B Fig**) and tested against Ddi antigen on the plate. Similarly, feces and urine from Mmo-vaccinated mice were tested for reactivity against Mmo in ELISA (**S8C** and **S8D Fig**, respectively). A secondary antibody that detects IgG, IgA and IgM was used. Urine was pooled per cage, so only two samples per group were analyzed. We could not detect any anti-PrP antibodies, excluding that the differences observed between vaccinated and control groups reflect assay interference rather than actual differences in prion shedding.

## Discussion

This is the first study which shows that vaccination reduces the shedding of CWD prions into urine and feces of infected animals. This is a very important finding for the future containment of CWD, both for farmed and wild-living animals. A vaccination that extends the lifetime of CWD-infected cervids without reducing the shedding of prions would be counter-productive, increasing the environmental load with prions over time and likely not breaking the disease transmission cycle. Our data show that despite vaccinated mice living longer, they still shed less prions early in disease course and may shed little or no additional prions during the additional time that they live with CWD.

CWD is a highly contagious prion disease that can be transmitted directly or indirectly between cervids. CWD-infected cervids contaminate the environment through continuous shedding of CWD prions in secretions such as feces, saliva, and urine, both in clinical and early pre-clinical stages [44–46]. Prions in the environment are very stable and can remain infectious over years. Experimental studies showed that prion seeding activity of cervid feces can remain detectable despite environmental factors such as desiccation and freeze-thaw cycles [42]. CWD prions effectively attach to certain soil types and are absorbed by plants, potentially contaminating the environment for decades [47–49], making disease containment and disease control very challenging. Horizontal transmission of CWD prions between cervids occurs through the alimentary tract that is the major route of entry for CWD prions [50]. From there, the process of neuro-invasion is slow and provides a window of opportunity for vaccination strategies, as antibodies do not have to cross the blood–brain barrier to be effective [29,30,51]. When the process of neuro-invasion is established, CWD prions egress from the central nervous system in a process not well understood and finally are found in peripheral tissues like muscles and secretions like urine, feces and saliva. Again, antibodies might have the opportunity to interfere in this process, acting outside the central nervous system [5,36,52].

Of note, prion infections do not evoke detectable immune responses. In contrast to traditional infectious agents that are 'foreign' to the immune system, prions use endogenous PrPC for their replication, which is a self-protein to the immune system. This also makes newly produced PrPSc a self-protein. The challenge for vaccine strategies is to effectively

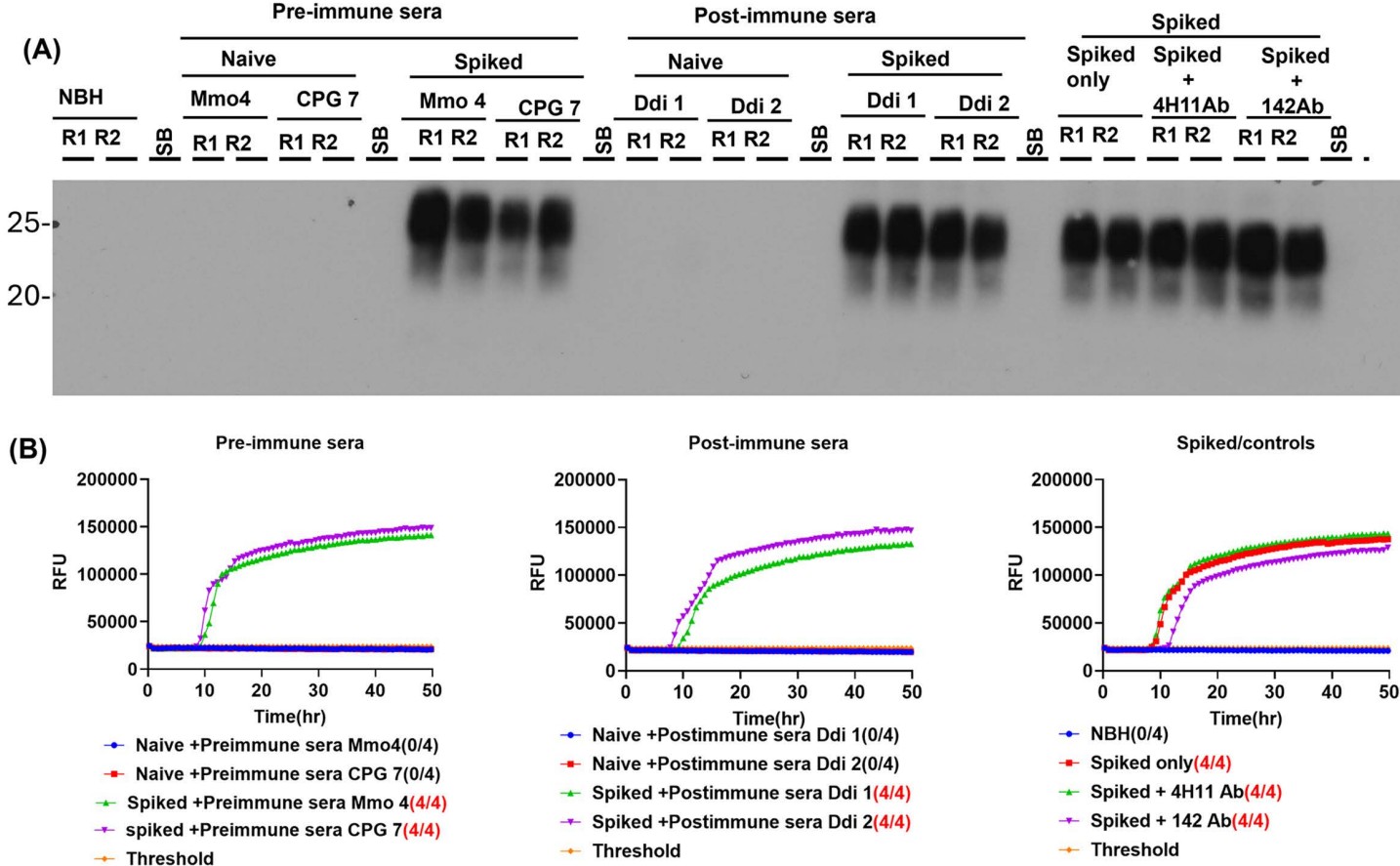

**Fig 6. Anti-PrP antibodies do not interfere with the IPR readout method.** Ten % fecal homogenates from naïve KI mice were treated with pre-immune mouse sera, post-immune mouse sera, mAb 4H11, or pAb 142 antibodies for 90 minutes followed by spiking 1:100 with brain homogenate of mouse adapted CWD, or without spiking (naïve), and all the samples were subjected to full IPR method. **(A)** Immunoblot showing PrPSc of PMCA reactions (in duplicate) pre-incubated with pre-immune sera (mouse Mmo4 and CpG7) or post-immune sera (mouse Ddi1 and Ddi2), either CWD spiked or not (naïve). Right shows spiked only reactions and spiked reactions pre-treated with mAb 4H11 or pAb 142 antibodies. SB, sample buffer. **(B)** RT-QuIC data showing the seeding activity in pre-treated and spiked samples from **A**. In both A and B, there was no detectable difference between anti-body treated and not treated samples.

overcome this self-tolerance but doing so without inducing unwanted side effects. We and others have shown experimentally that this is possible [27–32,34,35]. We very carefully analyzed vaccinated mice for potential side effects by weekly clinical assessment and weekly weighing. Pathology and histopathology analysis of harvested mice did not show abnormalities in spleen, kidney and brain, although some lymph node enlargement is found sometimes. In Ddi-vaccinated cervids, complete blood count parameters and immune cell profiles were normal.

Conceptually, there are two targets for active vaccination against prion diseases: PrPC, the substrate needed for prion conversion, or PrPSc that acts as template [5,36] In both situations, binding of anti-PrP antibodies might be sufficient to interfere by sterical hindrance with prion propagation, a process that involves a fitted protein-protein interaction. The PrPSc-targeted approach uses disease-specific epitopes (DSE), which are not present in PrPC and only are accessible in forming or already formed PrPSc. Several DSEs have been described: YYR, YML, and RL. The rigid loop region (RL) elicits systemic and mucosal immune responses when administered orally to white-tailed deer [53]. However, experimental trials in large animals using a YYR-based vaccine are inconsistent. In a sheep model challenged orally, the vaccine delayed the

onset of disease [54], but the disease was accelerated in elk when they were exposed to a CWD-contaminated environment for a long period [55]. Another group used a VPrP$^{Sc}$ vaccine, recombinant antigens that mimic a 4-rung beta solenoid fold of PrP$^{Sc}$, which delayed the onset of symptoms in a transgenic mouse model of a genetic human prion disease [56]. However, no studies have been reported with this vaccine in CWD models.

The PrP$^C$-targeting approach developed by us results in humoral and cellular immune responses against the immunogen without unwanted side effects [31,33,34]. Induced self-antibodies target surface-located PrP$^C$, which leads to depletion of PrP$^C$ and in blocking its role as a substrate for prion conversion [57,58]. This was shown in mouse and cervid models [31,33]. Using transgenic mice (Tg) expressing cervid PrP we have shown that vaccination prolongs the survival time by up to 70% upon intraperitoneal CWD challenge [35]. Unfortunately, such trangenic mice donot completely recapitulate the pathogenesis as found in the real cervid host. For instance, they do not shed prions into urine and feces [5,59]. To overcome this problem, we used gene-targeted KI mice in this study, for which it was shown that they closely recapitulate CWD pathogenesis, including shedding [38]. This allows us to investigate whether vaccination affects prion shedding of CWD prions, which was not feasible in transgenic mouse models of CWD infection. We first wanted to see whether vaccination targeting PrP$^C$ blocks the conversion of PrP$^C$ to PrP$^{Sc}$ outside the brain and thereby delays CWD neuro-invasion. Indeed, when we tested pre-clinical mice at comparable time points, we found lower amounts of PrP$^{Sc}$ in both spinal cord and brain tissues, when compared to the CpG only group.

We next were interested whether vaccination affects extra-CNS prion replication and the anterograde transport of CWD prions from the CNS to peripheral sites of the body, an aspect never analyzed before. Reduction of shedding of CWD prions from infected animals by vaccination is a key requirement for making vaccines that can contain CWD in the long term. The process of prion lateralization is not well understood, and it is unclear whether transport is occurring of PrP$^{Sc}$ preformed in the CNS, or whether generation of newly produced prions outside the brain is involved, or both. The latter would make this process a likely target for vaccine-induced anti-PrP antibodies targeting PrP$^C$ mostly. To test whether vaccine-generated antibodies affect CWD prion shedding and likely block prion propagation at peripheral sites of the body, we examined the amounts of PrP$^{Sc}$ in feces and urine of CWD-infected and previously vaccinated KI mice, in comparison to CpG control mice, using ultra-sensitive prion amplification methods. To evade the inhibitory effects of fecal debris and false positive results that were shown in previous studies [40,46,60], we combined IOME extraction with PMCA followed by RT-QuIC to increase both sensitivity and specificity of the IPR assay. IOME was included because it has been shown in previous studies that it enables the detection of low concentrations of PrP$^{Sc}$ in biological samples [43,61,62].

The interesting new finding of our study is that vaccination with both vaccines (Ddi and Mmo) reduced the shedding of CWD prions in feces and urine. For fecal samples, both the number of positive samples and the amount of seeding activity in samples still testing positive were reduced, when compared to samples taken from CpG-only treated mice. We analyzed fecal samples taken at 200, 250, 300 and 350 dpi, representing about 40%-70% of the incubation time to clinical disease in the used model. For assessing prion conversion activity in RT-QuIC, the final readout in IPR, we used established criteria like end-point dilution, time to threshold, maximum of range and area under the curve. Consistently, samples of Ddi or Mmo-vaccinated mice scored significantly lower in these criteria, indicating markedly less prion conversion activity and less CWD prions in these samples when compared to CpG control mice. Overall, best effects were found for Ddi-vaccinated mice, which aligns with our earlier studies that showed the self-antibodies produced by Ddi have a stronger anti-prion effect in cell culture neutralization than Mmo sera (46.2% vs. 5.3%), although titers in ELISA were higher for Mmo-vaccinated mice and cervids [31].

One caveat of our vaccine study is that we vaccinated systemically. Wildlife vaccination most likely will need an oral vaccination approach, due to feasibility issues and for generating mucosal immunity that likely is needed to protect against oral CWD infection. Using mouse models for validating oral vaccination and for translating results to the natural host will be complicated by the fact that the GI systems of mice and cervids are very different. Compared with rodents, the ungulate stomach milieu comprises four compartments that feature different retention and passage times and are associated

with varying pH conditions. Delivering vaccines orally in ruminants faces therefore significant challenges, including stability and degradation, limited mucosal permeability, and inefficient uptake by antigen-presenting cells. We have addressed this by using poly lactic co-glycolic acid (PLGA) nanoparticles that co-encapsulate vaccine and adjuvant [67]. Studies in mice showed that this approach can induce both mucosal and systemic immune responses against PrP$^C$, studies in cervids are ongoing.

In our study, prion shedding in feces in CWD-infected KI mice was detectable as early as 200 dpi, the earliest time point we had analyzed here. This matches findings in large animal models, in one elk study as early as 14 dpi [46], and 6 months post-infection in white-tailed deer, elk, and mule deer studies [40,41,44]. Prion shedding was very consistent throughout testing in the asymptomatic stages in the CpG control group, which matches previous observations in three cervid species [41]. Shedding is very critical for environmental CWD transmission and long-term persistence of CWD. Of note, in the Ddi-immunized group mice did not shed at various time points (e.g., mice numbers 1, 5 & 7 at 200–300 dpi) or shed only intermittently (mice numbers 3 & 8). The Ddi vaccine candidate is therefore promising for decreasing the CWD contamination in the environment over time.

Findings in urine were even more remarkable. Here, we tested early and very late time points, 150 and 450 days, respectively, corresponding to 30% and >90% of incubation time, with all mice still preclinical. Although we could test urine samples only when pooled and not from individual animals, our findings overall match what we found for fecal samples. The situation for 150 and 250 dpi was almost identical, with no detectable prion shedding for samples from both Ddi and Mmo-vaccinated mice, whereas 4 serial dilutions of urine tested positive in RT-QuIC after PMCA for CpG-only treated mice. For 450 dpi samples, all were negative for Ddi-vaccinated mice (3 animals), whereas Mmo mouse samples were positive for 4 dilutions (2 mice), and the CpG-only samples (1 mouse) was positive for 5 serial dilutions in RT-QuIC after PMCA amplification. In summary, no detectable prion conversion activity was found in samples coming from Ddi-vaccinated mice, Mmo-vaccinated mice shed at 450 dpi, and CpG-only treated mice shed at all analyzed time points, with 4–5 log higher amplification titers in the latter when compared to Ddi mice. Of note, our IPR readout method combines purification and amplification by PMCA with a re-amplification by RT-QuIC. This is necessary because rodent urine and feces contain only very small amounts of PrP$^{Sc}$ and could mean that very small differences in initial PrP$^{Sc}$ quantities could be enormously amplified in a non-linear way. The pre-clinical detection of CWD in urine aligned with our earlier findings in deer, at 13 and 16 months post infection (mpi) [63], and others, at 3 mpi in deer [11] and 18 mpi in white-tailed deer [44] until euthanasia at 66 mpi. Several studies in cervids demonstrated that CWD prion concentrations are higher and more consistently detectable in feces than in urine [11,42,44,60,64]. We observed detectable prion amounts in both feces and urine samples, using a KI mouse model of CWD infection. Pooled urine samples from the CpG control group showed seeding activity until the $10^{-5}$ dilution at 450 dpi (**Fig 5E**), and down to the $10^{-7}$ dilution at 350 dpi in pooled fecal samples (**Fig 3**) in RT-QuIC after PMCA, which indicates rather high amounts of CWD in feces and urine in this KI mouse model, and that the IPR technology described here is both sensitive and specific. Although others had mentioned that it is not necessary to do PMCA plus RT-QuIC after IOME extraction for urine samples of cervids [44], this was not the case for our study. For example, IOME with RT-QuIC detected prion seeding activity in pooled urine of the CpG control group only, but when accompanied by PMCA, it detected seeding activity in samples from both the Mmo and CpG groups (**S7 Fig**), indicating a higher sensitivity. Importantly, we also validated that the potential presence of anti-PrP antibodies in feces or urine would not interfere with the IPR readout method, even when testing the effect of highly concentrated experimental monoclonal and polyclonal antibodies. From this we conclude that the differences in prion shedding observed between vaccinated and control groups do not reflect assay interference but rather actual differences in prion shedding.

In summary, this is the first study to test the efficacy of active vaccination on CWD shedding. Our findings in a relevant knock-in mouse model of CWD infection are promising, showing that our vaccine candidates decrease the shedding of CWD prions in urine and feces. Whether this has the potential to break the CWD transmission cycle in the long term needs to be determined in vaccine studies in deer and elk, some of them ongoing. Taken together, the vaccination

strategy described here has obviously two additive effects. It has the potential to improve individual survival, which will hopefully translate into population effects. Importantly, it also reduces prion shedding, likely translating into reduction of CWD prions in the environment in the long term.

## Materials and methods

### Ethical statement

Four- to eight-week-old female KI mice expressing wild-type cervid PrP were used [37]. All animal experiments were approved by the University of Calgary Health Sciences Animal Care Committee (protocol# AC18–0138 and AC22–0106), following the guidelines issued by the Canadian Council on Animal Care.

### CWD prion material

Brain material from a CWD-infected captive white-tailed deer (WTD) was used to orally infect reindeer experimentally [65]. This reindeer developed clinical signs of prion infection and the brain was positive in WB and ELISA. Brain homogenate (BH) of the infected reindeer was used to challenge KI mice intraperitoneally [37], leading to a 100% attack rate. We used this KI mouse-adapted reindeer CWD in our study. 1% BH in PBS was used as inoculum.

### Mouse bioassay

We used three groups of mice (n = 8 per group): Ddi or Mmo vaccinated groups and a CpG control group. The first bleeding was done immediately before starting the vaccination procedure and used as pre-immune sera. The vaccine groups were subcutaneously (s.c.) injected with a priming dose of 100 µg of protein plus 5 µM CpG as adjuvant, followed by four booster doses consisting of 50 µg protein plus 5 µM CpG. The control group received 5 µM CpG in PBS only. Post-immune sera were collected 10 days after the fourth booster dose, then all groups were challenged intraperitoneally (i.p.) with 1% BH KI mouse-adapted reindeer CWD (Fig 1A).

### Feces and urine collection and processing

Individual or pooled feces and pooled urine were collected every 50 days post-inoculation (dpi) (Fig 1B). Regarding collection of urine, we did not use metabolic cages. However, we restrained the mouse in a tube and then placed the rear part of the body above a sterile petri dish and collected the urine using sterile tips to avoid any external contamination. The urine was collected individually and pooled before analysis because the volume obtained from each individual mouse was very minuscule. Feces were homogenized in sterile 1 × PBS 10% (w/v) homogenates in FastPrep-24 Lysing D Matrix tubes with ceramic beads (MP Biomedicals) using the FastPrep-24 Tissue Homogenizer (MP Biomedicals). Fecal homogenates were centrifuged twice at 18000 x g for 10 minutes to remove debris and were aliquoted and stored at −80°C until further use. Superparamagnetic iron oxide beads (IOME) [≈9 µm; BM547 Bangs Laboratories, Indiana] were used to concentrate PrP$^{Sc}$ as previously described [43]. Briefly, 2 µl of IOME beads were washed once with 500 µl PBS using a magnetic particle separator (MPS) (Pure Biotech, New Jersey), and supernatant was removed while the tubes were in the MPS. 500 µl of 10% fecal homogenate were added, and samples were placed on the end-over-end rocker for 2 hours at room temperature. Then, tubes were placed in MPS for 5 minutes, supernatant discarded, and beads were resuspended in 10 µl of 0.1% sodium dodecyl sulphate (SDS) in PBS and samples stored at −80°C until further use.

### Protein misfolding cyclic amplification assay (PMCA)

PMCA was performed using the protocol published by Arifin et al. [37] with some modifications. PMCA conversion buffer was prepared, which contains 4 mM EDTA, 1% Triton X-100 and 1 tablet cOmplete Protease Inhibitor Mini (Roche) in 1 × PBS. For the PMCA substrate, in a prion-free area, brains from naïve KI mice were collected and homogenized as

10% (w/v) BH using a Potter–Elvehjem PTFE pestle and glass tube (Sigma-Aldrich, #P7984) in ice-cold PMCA conversion buffer. The brain homogenates were aliquoted into 500 µL in sterile Eppendorf tubes and stored at −80 °C until use. In 0.2 mL tubes (ThermoFisher, #AB0337), three PTFE balls (McMaster-Carr, #9660K12, 3/32 in ø) and 90 µL of substrate were added. Moving to the prion area, the water bath (CC304-B, Huber) was set at 36.5 °C. The extracted 10% fecal homogenate or urine sample was diluted 1:10 in PMCA conversion buffer. Ten µL of each seed dilution was added to the 0.2 mL tubes, and all reactions were conducted in duplicate. Finally, PMCA tubes were sealed with parafilm, decontaminated in a 2.5% bleach for 5 min, rinsed with $H_2O$, and placed in a tube rack inside a microplate sonicator horn (431MPXH and Q700, QSonica) connected to the circulating water bath. Each cycle of PMCA is conducted for 24 hours which equals 48 cycles of 30 s sonication at 375–395 W followed by 29.5 min rest. For serial PMCA (sPMCA), 10 µL of PMCA product from the first round was transferred into 90 µL of fresh PMCA substrate and run for another 24 h with the same settings. This process was repeated to obtain three rounds of sPMCA with a total of 144 cycles. The PMCA products of the third round were used as a seed in the RT-QuIC assay.

## Real-time quaking induced conversion (RT-QuIC) assay

As readout of the PMCA products for fecal homogenates after IOME RT-QuIC was performed as previously described, using recombinant mouse PrP (aa23–231) as the substrate [63]. The reaction mixture included 10 µg of recombinant PrP, 10 µM Thioflavin-T (ThT), 170 mM NaCl, 1 × PBS (containing 130 mM NaCl), 1 mM EDTA, and water. Each well of a 96-well plate (Millipore Sigma, Ref# 3603) received 98 µL of this mixture plus 2 µL of the seed (PMCA products of 10% fecal homogenate extracted using IOME), tenfold serially diluted in RT-QuIC seed dilution buffer in quadruplicate. Plates were sealed, incubated at 42°C, and shaken at 700 rpm in a BMG Labtech FLUOstar plate reader for 50 hours. Thioflavin T fluorescence was measured every 15 minutes. A sample was considered positive if two out of the four replicates exceeded the threshold, defined as average RFU of the negative control plus five standard deviations.

## Preparation of the immunogen

Recombinant monomeric mouse (Mmo) and dimeric deer (Ddi) prion proteins were prepared as described previously [31,34]. Mmo consists of N-terminally his-tagged PrP encompassing amino acids (aa) 23–231 of mouse PrP, not containing the N- and C-terminal peptides (aa 1–22 and 232–254, respectively). The Ddi encoding constructs were synthesized by GeneArt and include two sequences of aa 23–231 of deer PrP, linked together by a short 7-amino acid linker (AGAIGGA). In brief, all constructs were cloned into the pQE30 expression vector (Qiagen) and expressed in E. coli BL21-Gold (DE3) pLysS cells (Stratagene). Guanidine hydrochloride 6M (pH 7.8) was used to lyse the bacterial cells and centrifuged at 6000 × g for 30 minutes to remove cell debris. Protein purification was done through Ni-affinity chromatography using Ni-NTA Superflow resin (Qiagen). The purified protein was eluted with the elution buffer and refolded using the dialysis buffer. Quality control tests were done on the produced protein, which included the BCA protein assay kit (Pierce, Thermo Scientific) to measure protein concentration, Coomassie blue staining of SDS-PAGE, Western blot and ELISA.

## Proteinase-K (PK) digestion

Brain homogenates (20%) or spinal cord homogenate (10%) prepared in 1X PBS were mixed with an equal volume of 2X lysis buffer and were digested with 50 µg/ml of PK at 37°C for one hour. Aliquots of the PMCA products of the feces and urine samples were mixed with 2x lysis buffer and digested with 50 µg/ml and 37 µg/ml of PK, respectively. Pefabloc protease inhibitor (1x; VWR) was added to stop the enzymatic reaction, followed by adding 3 x SDS sample loading buffer and boiling for 10 minutes at 95°C.

## SDS-PAGE and Western blot

Samples were separated on a 12.5% SDS-poly-acrylamide gel and transferred to PVDF membranes (Amersham, GE Healthcare). Membranes were blocked with 5% non-fat milk in Tris-buffered saline with a final concentration of 0.1% Tween-20 (TBST) for 1hr at room temperature. Membranes were probed with the anti-PrP monoclonal antibody 4H11 (1:500 or 1:700) [66] followed by washing with TBST. Horseradish peroxidase-conjugated goat anti-mouse IgG antibody (Sigma) was used as secondary antibody (1:5000), followed by washing with TBST. Luminita horseradish peroxidase substrate (MilliporeSigma) was used for developing. Images were acquired on X-ray films (Denville Scientific). Image J software was used to quantify and determine the relative values of PrP$^{Sc}$ signals. Calculations were done on Microsoft Excel, and graphs generated using GraphPad Prism 10.

## ELISA

ELISA was conducted as described previously [34]. In brief, 1 µg of either Ddi or Mmo recombinant protein in sodium-carbonate buffer (pH 9.5) was used to coat high-binding 96-well plates (Greiner Bio-One GmbH-Frickenhausen-Germany) overnight at room temperature. Plates were washed with PBS-T, then blocked using 3% bovine serum albumin (BSA) in PBS-T for 2 h at 37 °C. For analysis of sera, pre- or post-immune sera from every mouse were diluted 1:100 or serially diluted in 3% BSA, added to the plates, incubated for 1 h, and plates washed with PBS-T. The secondary antibody used was HRP-labeled anti-mouse IgG antibody (Jackson ImmunoResearch, West Grove, PA) diluted 1:4000 in 3% BSA. Signal detection was performed using the ABTS peroxidase substrate system (Kirkegaard & Perry Laboratories). Analysis of feces and urine was done as described recently [67]. Feces and urine of Ddi and Mmo-vaccinated mice were diluted 1:50 or 1:10 (w/v), respectively, and tested on ELISA plates coated with Ddi or Mmo. Goat anti-mouse recognizing IgG, IgA and IgM was used as secondary antibody at dilution of 1:3,000. Urine was pooled per cage, so only two sample per group were analyzed, but fecal samples were tested from individual mice (Ddi n = 7; Mmo n = 5; CPG n = 5). To assess the ELISA signal, the optical density (OD) was measured at 405 nm using a BioTek Synergy HT microplate reader.

## Epitope mapping

We tested 14 peptides covering the full-length mature cervid PrP, with each peptide consisting of 20 aa peptides with 5 aa overlaps (S1 Table). Peptide 6a contains the 3F4 epitope while peptide 6b represents the corresponding murine sequence. We used the same protocol as described by us previously [31,34]. Using CovaLink NH microtiter plates, activation of the wells was done using DSS bifunctional linker in carbonate buffer. Coating was done with 10 µg of each peptide or 1 µg of recombinant protein (control) at room temperature overnight. After washing, wells were blocked with BSA for 30 minutes at 37 °C and incubated with post-immune sera (1:100 in blocking buffer) for 2hr at 37 °C. Plate were washed and incubated with secondary antibodies (1:4000 in blocking buffer) for 1hr at 37 °C. After final washing, wells were incubated with ABTS substrate, the optical density was measured at 405 nm.

## Treatment of feces with anti-PrP antibodies followed by CWD spiking and IPR readout

Fecal homogenates (10% w/v) of naïve KI mice were treated with pre-immune (mouse Mmo#4 and CPG #7) or post-immune sera of vaccinated KI mice (mouse Ddi #1 and Ddi #2, which have low and high OD in ELISA, respectively) for 90 minutes, followed by spiking with mouse-adapted reindeer CWD brain homogenate at 1:100 dilution. IOME enrichment was applied for all samples, followed by three rounds of PMCA in duplicate. Normal brain homogenate (NBH) was used as negative control. Naïve feces were also treated with anti-PrP mAb 4H11 or pAb 142 (1:500 dilution) for 90 minutes and spiked as above and analyzed in IPR. PMCA products were digested with 50 µg/ml PK for 1 hr at 37oC and probed with an anti-PrP mAb in immunoblot analysis. RT-QuIC was used for testing the seeding activity of PMCA products and for completing the IPR readout method.

## Statistics

Statistical analyses were done using GraphPad Prism software (GraphPad 10.4.1, Software, USA). For statistical analysis of immunoblot signals, two-tailed independent Student's t-test groups or one-way analysis of variance (ANOVA), followed by Turkey's multiple comparison test, as applicable, were used. The area under the curve (AUC), time to threshold, and maximum of range were calculated using Omega data analysis (MARS version 4.01 R2). Statistical significance was tested using one-way analysis of variance (ANOVA) followed by Turkey's multiple comparison test, Values are expressed as mean ± SEM. Significance = $*p \leq 0.05$, $**p \leq 0.01$, and $***p \leq 0.001$.

## Supporting information

**S1 Fig. Linear epitope mapping of sera from Ddi and Mmo vaccinated mice.** The y-axis represents the optical density (OD) at 405 nm in ELISA, indicating the reactivity of post-immune sera from either Ddi (A) or Mmo (B) immunized mice against linear epitopes of deer PrP on the x-axis. The dashed horizontal line represents the cut-off, which is 3 times the average of preimmune sera. The data in panel (A) represents the average of Ddi sera from mice 1, 3, 4 and 6. The data in panel (B) represents the average of Mmo sera from mice 1, 2, 3 and 6. Data are presented as mean ± SD of results from four individual mice of each group. The amino acid sequences for linear epitopes are shown in S1 Table.
(PDF)

**S2 Fig. RT-QuIC data showing the seeding activity in feces from vaccinated and control mice at 250 dpi.** Fecal homogenates from individual mice were taken at 250 dpi and extracted using IOME followed by three rounds of PMCA reactions seeded with 10–1 dilution of 10% fecal homogenates and performed in duplicates. Positive control for PMCA was naïve feces spiked with mouse-adapted CWD and negative control was naïve feces. PMCA products were analyzed using RT-QuIC at 10–2 dilution. **(A-C),** representative RT-QuIC graphs showing the seeding activity in feces from vaccinated or CpG control mice. Samples were considered positive when 2 out of 4 wells crossed the threshold, which is defined as the average RFU of the negative control group plus five times its standard deviation. The y-axis represents the RFU, and the x-axis represents the time in hours (hr). **(D)** Chi square test, **(E)** time to threshold, **(F)** maximum of range, and **(G)** area under curve. Graphs were generated using GraphPad Prism (version 10). Statistical analysis was done using Chi-square test with **** p-value < 0.0001, or One-way ANOVA followed by a Tukey's multiple comparison. ns means not significant.
(PDF)

**S3 Fig. Seeding activity in feces from vaccinated and control mice at 300 dpi.** Fecal homogenates from individual mice taken at 300 dpi were extracted and subjected to IPR analysis. IOME extraction was followed by three rounds of PMCA reactions seeded with 10–1 dilution of 10% fecal homogenates. Positive control for PMCA was naïve feces spiked with mouse-adapted CWD and negative control was naïve feces. All samples and controls were subjected to PMCA and products analyzed using RT-QuIC at 10–1 dilution. **(A-C),** representative RT-QuIC graphs showing the seeding activity in feces from vaccinated or control mice. Samples were considered positive when 2 out of 4 wells crossed the threshold, which is defined as the average RFU of the negative control group plus five times its standard deviation. The y-axis represents the RFU, and the x-axis represents the time in hours (hr). **(D)** Chi square test, **(E)** time to threshold, **(F)** maximum of range, and **(G)** area under curve. Graphs were generated using GraphPad Prism (version 10). Statistical analysis done using Chi-square test with **** p-value < 0.0001 or One-way ANOVA followed by a Tukey's multiple comparison for time to threshold **(E)**: Ddi vs. Mmo ** p-value = 0.0043, Ddi vs. CpG ** p-value = 0.0037; for maximum of range **(F)**: Ddi vs. Mmo * p-value = 0.0498, Ddi vs. CpG * p-value = 0.0149. ns: not significant; and for area under curve **(G)**: Ddi vs. Mmo * p-value = 0.0225, Ddi vs. CpG * p-value = 0. 0120.
(PDF)

**S4 Fig. Seeding activity in feces from vaccinated and control mice at 350 dpi.** Fecal samples taken from individual mice at 350 dpi were extracted using IOME and subjected to three rounds of PMCA. Positive control for the PMCA was naïve feces spiked with mouse-adapted CWD and negative control was naïve feces. All samples and controls were subjected to PMCA and products analyzed using RT-QuIC at 10–5 dilution. **(A-C),** representative RT-QuIC graphs showing the seeding activity in feces from vaccinated or control mice. Samples were considered positive when 2 out of 4 wells crossed the threshold, which is defined as the average RFU of the negative control group plus five times its standard deviation. The y-axis represents the RFU, and the x-axis represents the time in hours (hr). **(D)** Chi square test, **(E)** time to threshold, **(F)** maximum of range, and **(G)** area under curve. Graphs were generated using GraphPad Prism (version 10). Statistical analysis was done using Chi-square test with **** $p$-value $< 0.0001$, or Two-way ANOVA followed by a Tukey's multiple comparison for area under curve **(G):** Ddi vs CpG * $p$-value $= 0.0453$ and Mmo vs CpG ** $p$-value $= 0.0052$; and for maximum of range **(F):** * $p$-value $= 0.0308$. (PDF)

**S5 Fig. RT-QuIC prion seeding activity in third round PMCA products for urine of mice vaccinated with Ddi or Mmo and control CpG group.** The graphs represent RT-QuIC results of serial dilutions (10–1 to 10–6) for 150 dpi **(A)** and 250 dpi **(B)** pooled urine samples from all 3 groups after three rounds of PMCA using mouse rPrP substrate. Fluorescence signals were measured every 15 min. The *x*-axis represents the reaction time (hour), the *y*-axis the relative fluorescence units (RFU). The threshold was based on the average fluorescence values of the negative control $+ 5 \times$ SD used in every assay. Each curve represents an average of 4 technical replicates. (PDF)

**S6 Fig. RT-QuIC data showing the CWD prion seeding activity in third round PMCA products for urine of mice at 450 dpi.** The graphs represent results of serially diluted RT-QuIC reactions (10-1 to 10—6) for urine samples taken at 450 dpi after three rounds of PMCA. Fluorescence signals were measured every 15 min. The x-axis represents the reaction time (hours), the y-axis represents the relative fluorescence units (RFU). The threshold was based on the average fluorescence values of the negative control $+ 5 \times$ SD used in every assay. Each curve represents the average of 4 technical replicates. (PDF)

**S7 Fig. RT-QuIC data showing the seeding activity in 450 dpi pooled urine samples extracted with IOME (A) or IOME followed by three rounds of PMCA (B).** Fluorescence signals were measured every 15 min. The *x*-axis represents the reaction time (hours), the *y*-axis represents the relative fluorescence units (RFU). The threshold was based on the average fluorescence values of the negative control $+ 5 \times$ SD used in every assay. Each curve represents an average of 4 technical replicates. (PDF)

**S8 Fig. No anti-PrP antibodies were detected in feces and urine of vaccinated mice.** ELISArepresenting the reactivity of **(A)** feces from Ddi vaccinated mice (300 dpi) diluted 1:50 (w/v), or**(B)** 1:10 diluted urine, was tested using Ddi antigen to coat the plate. **(C)** represents the reactivityof feces from Mmo-vaccinated mice (300 dpi) diluted 1:50 (w/v) and **(D)** represents 1:10 dilutedurine against Mmo as antigen used to coat the plate. For all panels goat anti-mouse detecting IgG, IgA and IgM was used as secondary antibody. Urine was pooled per cage, so only two samples per group were analyzed. (PDF)

**S1 Table. Sequence of deer PrP peptides used in epitope mapping.** (PDF)

**S2 Table. Number of mice per group for pooled urine analysis.** (PDF)

**S3 Table. PrP$^C$ antibody titer and positive replicates in CWD seeding activity in feces at different time points.** (PDF)

## Acknowledgments

In-kind contributions were provided by the Canadian Food Inspection Agency, The Canadian Agri-Food Policy Institute, National Boreal Caribou Knowledge Consortium, and Métis Nation of Alberta. We acknowledge support (contributed with materials and manpower/time commitments) from Alberta Conservation Association, Office of the Chief Scientist, Alberta Environment and Protected Areas, Saskatchewan Ministry of Environment, and Parks Canada Agency.

## Author contributions

**Conceptualization:** Hanaa Ahmed-Hassan, Chimoné S. Dalton, Sabine Gilch, Hermann M. Schätzl.

**Data curation:** Hanaa Ahmed-Hassan, Dalia Abdelaziz, Yo-Ching Cheng, Kevin Low, Shirley Phan, Byron Kruger, Chimoné S. Dalton, Sabine Gilch, Hermann M. Schätzl.

**Formal analysis:** Hanaa Ahmed-Hassan, Dalia Abdelaziz, Yo-Ching Cheng, Kevin Low, Shirley Phan, Byron Kruger, Chimoné S. Dalton.

**Funding acquisition:** Sabine Gilch, Hermann M. Schätzl.

**Investigation:** Hanaa Ahmed-Hassan, Dalia Abdelaziz, Yo-Ching Cheng, Kevin Low, Shirley Phan, Byron Kruger, Chimoné S. Dalton.

**Methodology:** Hanaa Ahmed-Hassan, Dalia Abdelaziz, Byron Kruger, Chimoné S. Dalton, Walker S. Jackson, Sabine Gilch.

**Project administration:** Hermann M. Schätzl.

**Resources:** Lech Kaczmarczyk, Walker S. Jackson, Sabine Gilch, Hermann M. Schätzl.

**Supervision:** Dalia Abdelaziz, Hermann M. Schätzl.

**Validation:** Hanaa Ahmed-Hassan, Dalia Abdelaziz, Yo-Ching Cheng, Chimoné S. Dalton, Hermann M. Schätzl.

**Visualization:** Hanaa Ahmed-Hassan, Dalia Abdelaziz, Hermann M. Schätzl.

**Writing – original draft:** Hanaa Ahmed-Hassan, Byron Kruger, Chimoné S. Dalton, Walker S. Jackson, Sabine Gilch, Hermann M. Schätzl.

**Writing – review & editing:** Hanaa Ahmed-Hassan, Sabine Gilch, Hermann M. Schätzl.

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
