## [Decision Letter · Decision Letter 0]

1 Feb 2026

PPATHOGENS-D-25-03279

Prion shedding is reduced by chronic wasting disease vaccination

PLOS Pathogens

Dear Dr. Schatzl,

Thank you for submitting your manuscript to PLOS Pathogens. After careful consideration, we feel that it has merit but does not fully meet PLOS Pathogens's publication criteria as it currently stands. Therefore, we invite you to submit a revised version of the manuscript that addresses the points raised during the review process.

We look forward to receiving your revised manuscript.

Kind regards,

Jason C Bartz

Academic Editor

PLOS Pathogens

Neil Mabbott

Section Editor

PLOS Pathogens

Sumita Bhaduri-McIntosh

Editor-in-Chief

PLOS Pathogens

orcid.org/0000-0003-2946-9497

Michael Malim

Editor-in-Chief

PLOS Pathogens

orcid.org/0000-0002-7699-2064

**Additional Editor Comments:**

Please be mindful of the comments of the three reviews and pay special attention to Review 2, major concern 1 about how the presence of anti-PrP antibodies in feces and urine may affect PMCA detection and Reviewer 3 on discussing the differences between murine and cervid GI tracks. Thank you.

**Journal Requirements:**

https://journals.plos.org/plospathogens/s/submission-guidelines#loc-parts-of-a-submission

- TM on pages: 12, and 13.

Potential Copyright Issues:

i) Figure 1A. Please confirm whether you drew the images / clip-art within the figure panels by hand. If you did not draw the images, please provide (a) a link to the source of the images or icons and their license / terms of use; or (b) written permission from the copyright holder to publish the images or icons under our CC BY 4.0 license. Alternatively, you may replace the images with open source alternatives. See these open source resources you may use to replace images / clip-art:

6) In the online submission form, you indicated that Correspondence and material requests should be addressed to Dr. Hermann Schatzl (hschaetz@ucalgary.ca).. All PLOS journals now require all data underlying the findings described in their manuscript to be freely available to other researchers, either

1. In a public repository

2. Within the manuscript itself

3. Uploaded as supplementary information.

7) Please amend your detailed Financial Disclosure statement. This is published with the article. It must therefore be completed in full sentences and contain the exact wording you wish to be published.

2) If any authors received a salary from any of your funders, please state which authors and which funders..

**Reviewers' Comments:**

Reviewer's Responses to Questions

**Part I - Summary**

Reviewer #1: To effectively control the ongoing spread of Chronic Wasting Disease (CWD) in cervids, halting lateral transmission is likely the most practical strategy. In this study, Schatzl's group evaluated the efficacy of vaccination in reducing prion shedding in the feces and urine of CWD-infected knock-in mice. Their findings indicate that vaccination significantly decreased CWD shedding in both feces and urine throughout a substantial portion of the incubation period. To my knowledge, this is the first study to demonstrate that a vaccine can effectively reduce CWD shedding, which is a significant advancement.

Reviewer #2: This manuscript presents a well-executed study investigating the effect of PrPC-targeted vaccination on CWD prion shedding in feces and urine using knock-in mice that recapitulate cervid CWD pathogenesis. The research addresses an important question: whether vaccination can reduce environmental contamination through decreased prion shedding, which is critical for CWD containment strategies. The authors demonstrate that vaccination with recombinant dimeric deer PrP (Ddi) and monomeric mouse PrP (Mmo) reduces CWD shedding in both feces and urine during preclinical stages of infection.

The study's strengths include: (1) the use of an appropriate knock-in mouse model that recapitulates CWD pathogenesis including shedding, (2) the development and application of a sensitive IPR (IOME-PMCA-RT-QuIC) detection technique, (3) systematic longitudinal sampling across multiple timepoints representing 30-90% of incubation time, and (4) the demonstration of reduced shedding alongside survival time.

However, the manuscript would benefit from several clarifications and modifications. There are concerns regarding potential overstatement of results given the ultra-sensitive amplification methods used, questions about whether residual antibodies in excreta might interfere with amplification assays, and the need for better explanation of methodological details. The discussion is overly lengthy with repetitive descriptions of results, and some statements about prion concentrations and log-reductions may be misleading given that measurements are based on amplified products rather than direct detection.

Overall, the study represents a reasonable demonstration that vaccination can reduce prion shedding in a prion disease model, which is a significant finding for CWD control strategies.

Reviewer #3: This manuscript reports significant decrease of prion shedding in a cervid PrP knock-in mouse model of chronic wasting disease after vaccination with heterologous monomeric mouse PrP or homologous dimeric cervid PrP. The researchers report high titers of antibodies against both targets for the respective vaccines, a 30-day delay in terminal disease and significant decrease in shedding in feces and urine throughout disease course. These findings are important because vaccination that extends life spans of CWD-infected animals without decreased shedding actually increases environmental contamination and possibly indirect spread of CWD. The experiments are well-designed, data are clear and well-presented, and conclusions well-supported. The authors should address the following comments in a revised manuscript.

**Part II – Major Issues: Key Experiments Required for Acceptance**

Please use this section to detail the key new experiments or modifications of existing experiments that should be absolutely required to validate study conclusions.required to validate study conclusions.

Reviewer #1: (No Response)

Reviewer #2: Major concern 1:

The authors should address whether residual antibodies in feces and urine samples, even at minimal levels, could interfere with the initial PMCA amplification step. Since direct detection of prions is not possible in these samples, the presence of vaccine-induced antibodies (either free or already bound to PrPres seeds) could affect amplification efficiency and lead to misinterpretation of results. This concern is particularly relevant because the differences observed between vaccinated and control groups could potentially reflect assay interference rather than actual differences in prion shedding.

The authors should discuss this possibility and, if feasible, provide experimental evidence that antibodies do not interfere with the IPR technique. This could include spiking experiments with known amounts of prions in the presence of immune sera at concentrations expected in excreta.

Major concern 2

Throughout the manuscript, the authors make statements that could be misinterpreted by readers regarding prion concentrations and fold-reductions (e.g., "5-log reduction," "high prion concentration," "10e-5 dilutions"). The read-out method combines purification/amplification by PMCA with re-amplification by RT-QuIC. This methodology, used because samples contain minuscule amounts of PrPres (seeds), means that extremely small differences in initial PrPres quantity are enormously amplified. Therefore, statements about "log reductions" could be misinterpreted as referring to direct measurements of seed quantities, which is not accurate.

Specific examples to revise:

• Lines referencing "a 5-log reduction of shedding in Ddi-vaccinated mice"

• "We observed high prion concentration in both feces and urine samples"

• "Pooled urine samples from the CpG control group showed seeding activity until the 10e-5 dilution at 450 dpi"

• "...whereas 4 serial dilutions of urine tested positive for CpG-only treated mice"

The authors must revise these statements throughout the manuscript to clarify that these refer to seeding activity in amplified products, not direct prion concentrations. They should avoid language that could mislead readers about the actual quantities of prions in the original samples.

Major concern 3

The authors should provide better justification and discussion of the necessity of using a coupled method involving two different amplification techniques (PMCA + RT-QuIC) and how this might influence interpretation of observed differences. The use of serial PMCA (3 rounds) followed by RT-QuIC introduces variability that is well-documented for PMCA. The discussion should address:

1. Why this coupled approach was necessary for this particular study

2. How the known variability of serial PMCA might affect the interpretation of differences between groups

3. Whether the differences observed could be influenced by the amplification process itself rather than solely reflecting differences in original prion content

Reviewer #3: 1. Does PrP decrease on peripheral cells (FDCs, B cells, eg) after vaccination, either transiently or during disease course, that might contribute to less prion replication and shedding? If so, any adverse side effects from decreased PrP expression?

**Part III – Minor Issues: Editorial and Data Presentation Modifications**

Reviewer #1: A key conclusion from the study is that vaccination delays neuro-invasion. While this conclusion is generally supported by the data, the graphs presented in Figures 1g, h, and i may not adequately illustrate this difference. The authors should consider employing alternative graph types, such as bar graphs, to more clearly depict the distinctions between vaccinated and unvaccinated subjects.

There appear to be considerable discrepancies among individual vaccinated animals. For instance, the results from two Ddi cages in Figure 3b show significant variation. The authors should provide an explanation for this variability.

Figure legends should be more informative, including details such as sample size (n). For example, Figures 2a and 2c lack explanations for the numbers displayed along the X-axis. Are these numbers representative of individual animals? Additionally, why are there no error bars?

Figure 4d presents a chi-square test, yet there is no indication of statistical differences.

The description in Figure 4e should include an explanation regarding the absence of statistical significance.

The discussion section could be shortened.

Reviewer #2: Minor issue 1

It would have been helpful if the text lines had been numbered to facilitate the review process.

Minor issue 2

In the introduction, it would be helpful to inform readers whether there is a differential shedding mechanism that might be influenced by the strain type used or the model type employed. In the discussion, the authors mention a cervid PrP expression model that does not exhibit shedding. This is important for clarifying that the model used, although a knock-in mouse whose physiology differs greatly from deer, might have shedding behavior similar to cervids. Please comment on this in the introduction to provide appropriate context.

Minor issue 3

Reference 37 does not appear to demonstrate shedding in urine. Please verify whether an additional reference should be added here, such as reference 39 which is cited in the discussion regarding urine shedding.

This same comment applies to the statement in the first results section: "We used gene-targeted KI mice where mouse PrP was replaced with cervid PrP, which recapitulate the pathogenesis of CWD, including shedding of CWD prions into feces and urine."

Minor issue 4

In the results section, the authors state: "although a statistical analysis is not possible." Please clarify why statistical analysis was not possible in this context.

Minor issue 5

The discussion section is excessively long and contains portions that repeat descriptions of results already presented. Please reduce the length of the discussion by eliminating redundant descriptions and focusing on interpretation and implications of the findings.

Minor issue 6

The statement "to correlate to ELISA titers to some extent" in the discussion does not accurately reflect what is shown in Supplementary Table 3. The table does not support a clear correlation. Please revise this statement to more accurately reflect the data presented.

Minor issue 7

Please discuss the changing percentages of shedding over time shown in Table 1. The variation in % shedding across timepoints (200, 250, 300, 350 dpi) within groups deserves interpretation.

Minor issue 8

At some point in the manuscript, it would be beneficial to clarify the previously observed advantages of Ddi versus Mmo vaccination, or vice versa, to help readers understand the rationale for using both immunogens.

Minor issue 9

The sample collection procedures, especially for urine, are not sufficiently detailed. Please specify:

• Is urine collected individually or pooled during collection?

• What measures were taken to avoid contamination if animals were housed in the same cage?

• Were metabolic cages used to collect urine?

• Please also explain the details of how urine samples were processed.

Minor issue 10

Panels d and f in Figure 1 are not explained in the figure legend. Please add appropriate descriptions.

Minor issue 11

In Figure 1, it does not make sense to discuss "delay" when the dpi timepoints at which samples were collected after euthanasia are different between animals. The PrPres signals are perfectly correlated with dpi. The statement in the figure legend "The delay in the neuro-invasion in the vaccinated groups represented in graph c and e" cannot be made unless all animals had been sacrificed at the same time.

Minor issue 12

How do the authors explain that Ddi and Mmo induce so few common antibodies? It is surprising that, given the characteristics of Ddi and Mmo, specific antibodies for each antigen can be induced so distinctly. It is particularly surprising that antibodies induced with Ddi recognize the Mmo antigen so poorly. Please clarify this in the discussion.

Minor issue 13

How does the 3-round PMCA + RT-QuIC approach compare to a potential alternative read-out consisting only of PMCA but with more rounds?

Minor issue 14

Related to Major concern 2 regarding overstatement and the re-amplification read-out method: Given the way of quantifying by RT-QuIC following prior amplification, it does not make sense to distinguish between values of 50, 75, and 100. This could be misinterpreted as suggesting that results of 75, 50 versus 100, 100 represent different outcomes when, with this procedure, both should simply be considered positive. Since the authors have already explained what is considered positive and negative, they should standardize by indicating positive or negative. Otherwise, this misleads the reader by appearing to differentiate between values of 50, 75, and 100.

Minor issue 15

In the urine study shown in Figure 5, were amplifications performed with additional rounds of PMCA beyond the three rounds used?

Reviewer #3: 2. Since you have IPR data on serial dilutions of excreta, consider estimating reduction in SD50 by Reed-Meuncsh or Spearman-Karber methods to estimate titer reduction after vaccination, or provide a compelling argument why not.

3. Consider discussing that, despite the mice living longer, they still shed less prions early in disease course and may shed little or no additional prions during the additional thirty days that they live with CWD.

4. GI systems of mice and cervids are very different. Briefly discuss this caveat when translating these results to the natural host.

5. Figure 1: Indicate in the legend “PrPRes levels in c and d, Brain, and e and f, Spinal cord" and define "SB".

6. Figure 5: Panels d and e hard to read

7. Update the last column in SF7 from "Titer" to "Reciprocal Titer" (suggested) or “Fold-Dilution”

PLOS authors have the option to publish the peer review history of their article (what does this mean?). If published, this will include your full peer review and any attached files.). If published, this will include your full peer review and any attached files.

.

Reviewer #1: No

Reviewer #2: No

Reviewer #3: No

**Figure resubmission:**
---

## [Decision Letter · Decision Letter 1]

10 Apr 2026

Dear Dr Schatzl,

We are pleased to inform you that your manuscript 'Prion shedding is reduced by chronic wasting disease vaccination' has been provisionally accepted for publication in PLOS Pathogens.

Best regards,

Jason C Bartz

Academic Editor

PLOS Pathogens

Neil Mabbott

Section Editor

PLOS Pathogens

Sumita Bhaduri-McIntosh

Editor-in-Chief

PLOS Pathogens

orcid.org/0000-0003-2946-9497

Michael Malim

Editor-in-Chief

PLOS Pathogens

orcid.org/0000-0002-7699-2064

Thank you for your thorough and thoughtful revisions.

Reviewer Comments (if any, and for reference):

Reviewer's Responses to Questions

**Part I - Summary**

Reviewer #1: (No Response)

Reviewer #2: The authors have addressed my concerns and the revised manuscript is clearly improved. In particular, the new spiking experiments and the additional ELISA analyses. The revised version also provides a better rationale for the combined IOME/PMCA/RT-QuIC approach, adds useful methodological clarifications, and improves the discussion of the translational caveats of the mouse model.

Overall, I consider that the revision has substantially improved the manuscript and that my concerns have been addressed to a satisfactory extent.

Reviewer #3: The authors have adequately addressed all concerns and I support publishing this manuscript.

**Part II – Major Issues: Key Experiments Required for Acceptance**

Please use this section to detail the key new experiments or modifications of existing experiments that should be absolutely required to validate study conclusions.required to validate study conclusions.

Reviewer #1: (No Response)

Reviewer #2: I do not think additional experiments are required.

Reviewer #3: (No Response)

**Part III – Minor Issues: Editorial and Data Presentation Modifications**

Reviewer #1: (No Response)

Reviewer #2: Everything was addressed.

Reviewer #3: (No Response)

PLOS authors have the option to publish the peer review history of their article (what does this mean?). If published, this will include your full peer review and any attached files.). If published, this will include your full peer review and any attached files.

.

Reviewer #1: No

Reviewer #2: No

Reviewer #3: No

---

## [Editor Report · Acceptance letter]

Dear Dr Schätzl,

We are delighted to inform you that your manuscript, "Prion shedding is reduced by chronic wasting disease vaccination," has been formally accepted for publication in PLOS Pathogens.

Best regards,

Sumita Bhaduri-McIntosh

Editor-in-Chief

PLOS Pathogens

orcid.org/0000-0003-2946-9497

Michael Malim

Editor-in-Chief

PLOS Pathogens

orcid.org/0000-0002-7699-2064